# Relationships of Pelvic Vein Diameter and Reflux with Clinical Manifestations of Pelvic Venous Disorder

**DOI:** 10.3390/diagnostics12010145

**Published:** 2022-01-07

**Authors:** Sergey Gavrilov, Anatoly Karalkin, Nadezhda Mishakina, Oksana Efremova, Anastasia Grishenkova

**Affiliations:** Savelyev University Surgical Clinic, Pirogov Russian National Research Medical University, 10/5 Leninsky Prospect, 119049 Moscow, Russia; olittosg@gmail.com (A.K.); mishakina.78@mail.ru (N.M.); lpl2@yandex.ru (O.E.); ngrishenkova@rambler.ru (A.G.)

**Keywords:** pelvic venous disorder, pelvic pain, pelvic vein diameter, pelvic venous reflux, duplex ultrasound, single-photon emission computed tomography of pelvic veins

## Abstract

The causes of chronic pelvic pain (CPP) in patients with pelvic venous disorder (PeVD) are not completely understood. Various authors consider dilation of pelvic veins (PeVs) and pelvic venous reflux (PVR) as the main mechanisms underlying symptomatic forms of PeVD. The aim of this study was to assess relationships of pelvic vein dilation and PVR with clinical manifestations of PeVD. This non-randomized comparative cohort study included 80 female patients with PeVD who were allocated into two groups with symptomatic (n = 42) and asymptomatic (n = 38) forms of the disease. All patients underwent duplex scanning and single-photon emission computed tomography (SPECT) of PeVs with in vivo labeled red blood cells (RBCs). The PeV diameters, the presence, duration and pattern of PVR in the pelvic veins, as well as the coefficient of pelvic venous congestion (C_PVC_) were assessed. Two groups did not differ significantly in pelvic vein diameters (gonadal veins (GVs): 7.7 ± 1.3 vs. 8.5 ± 0.5 mm; parametrial veins (PVs): 9.8 ± 0.9 vs. 9.5 ± 0.9 mm; and uterine veins (UVs): 5.6 ± 0.2 vs. 5.5 ± 0.6 mm). Despite this, C_PVC_ was significantly higher in symptomatic versus asymptomatic patients (1.9 ± 0.4 vs. 0.7 ± 0.2, respectively; *p* = 0.008). Symptomatic patients had type II or III PVR, while asymptomatic patients had type I PVR. The reflux duration was found to be significantly greater in symptomatic versus asymptomatic patients (median and interquartile range: 4.0 [3.0; 5.0] vs. 1.0 [0; 2.0] s for GVs, *p* = 0.008; 4.0 [3.0; 5.0] vs. 1.1 [1.0; 2.0] s for PVs, *p* = 0.007; and 2.0 [2.0; 3.0] vs. 1.0 [1.0; 2.0] s for UVs, *p* = 0.04). Linear correlation analysis revealed a strong positive relationship (Pearson’s r = 0.78; *p* = 0.007) of CPP with the PVR duration but not with vein diameter. The grade of PeV dilation may not be a determining factor in CPP development in patients with PeVD. The presence and duration of reflux in the pelvic veins were found to be predictors of the development of symptomatic PeVD.

## 1. Introduction

The dilation of pelvic (parametrial, uterine and gonadal) veins and reflux in them are associated with the presence of chronic pelvic pain (CPP) in 60–76% of patients with pelvic venous disease (PeVD) [1,2,3,4,5]. According to the UIP consensus document, the dilation of pelvic veins (PeVs) is defined as an increase in their diameter of greater than 5 mm, and the pelvic venous reflux (PVR) is considered pathological if it lasts for greater than 1 s [2]. PVR of greater than 2 s in several pelvic venous collecting pools was found to be associated with severe CPP [4]. Some authors [6] also pointed out a relationship between the degree of PeV dilation and clinical manifestations of PeVD. Ganeshan et al. reported that PeV dilation of greater than 8 mm can be a criterion for the diagnosis of PeVD [6]. However, a study by Dos Santos et al. has demonstrated that the diameter of ovarian veins is not a predictor of reflux in them [7]. Whiteley et al. have repeatedly pointed out that the presence of reflux and its duration are major determinants of the severity of the disease and serve as an indication for intervention on PeVs [8,9,10,11]. However, in 2021, Szary et al. presented a classification of ovarian veins system insufficiency based on the degree of their dilatation [12]. These contradictory opinions indicate the need for further research to identify factors determining clinical manifestations of PeVD, primarily CPP. The present study was aimed at investigating relationships between pelvic vein dilation, PVR, and the presence and severity of clinical manifestations of PeVD.

## 2. Materials and Methods

This non-randomized comparative cohort study included 80 female patients aged 23 to 42 years (interquartile range: 9 years) with PeVD. Eligible for inclusion were women aged from 18 to 45 years with pelvic varicose veins diagnosed by DUS and without post-thrombotic syndrome (PTS), pregnancy, virgo intacta status (a contraindication for transvaginal DUS), as well as disorders that can be accompanied by CPP. The study was approved by the local Ethics Committee of the University, and all patients signed an informed consent form before entering the study.

The patients were allocated into two groups based on the results of clinical examination, DUS of the pelvic veins and consultations of related specialists (gynecologist, urologist and neurologist). The first group (n = 42) included patients with symptoms and signs of PeVD (CPP, heaviness in hypogastrium, dyspareunia, dysuria, vulvar varicosities). The intensity of pain syndrome was evaluated using a visual analogue scale (VAS) ranging from 0 to 10 scores (where 0 is no pain and 10 is maximum pain) and was graded as mild (1–4), moderate (5–6) or severe (7–10). The second group (n = 38) consisted of patients without clinical manifestations of PeVD, in whom pelvic varicose veins were identified accidentally during a routine gynecological ultrasound examination of the pelvic organs or during examination for chronic venous disease (CVD) of the lower extremities. All patients underwent DUS and single-photon emission computed tomography (SPECT) of the pelvic veins with in vivo labeled red blood cells (RBCs). The study design is shown in Figure 1.

### 2.1. Assessment of the Pelvic Veins

*Ultrasound examinations* of the pelvic and lower limb veins were performed using Esaote MyLab Class C (Esaote, Genova, Italy) devices. The examination protocol included a sequential assessment of the status of superficial and deep veins of the lower extremities, perineal veins, pelvic (parametrial, uterine, gonadal) veins, iliac veins, inferior vena cava and renal veins. During the ultrasound studies, breathing (Valsalva maneuver) and compression tests were used to detect blood reflux in the veins of the lower extremities, perineum and pelvis. An increase in the diameter of the pelvic veins (i.e., only parametrial, uterine and gonadal veins) of more than 5 mm was considered pathological [2]. Reflux was defined as retrograde flow lasting for more than 1 s in PeVs and deep veins of the lower extremities [2,5].

DUS of the veins of the lower extremities and perineal veins was performed by standard protocol using a 7.5–13 MHz transducer in the supine and standing position of the patient and included an assessment of the vein patency and diameter, the presence of pathological reflux in the sapheno-femoral and sapheno-popliteal junctions, trunks of the great (GSV) and small (SSV) saphenous veins, and veins of the labia majora (vulvar veins), as well as the presence of retrograde blood flow in the ostial tributaries of the GSV (superficial epigastric, superficial circumflex iliac, superficial external pudendal veins).

Transabdominal duplex ultrasound (TADUS) of the iliac veins, inferior vena cava, renal and gonadal veins was performed using convex (3–5 MHz) and linear (3–18 MHz) transducers in the patient’s supine and half-sitting (with trunk raised to 45°) positions. The patency and diameters of the external, internal and common iliac veins, inferior vena cava and renal veins were assessed. The maximum blood flow velocity in the iliac and left renal veins was measured, and the presence and duration of reflux in the internal iliac vein (IIV) and GVs were assessed. The left renal vein status was studied using transverse and longitudinal scanning of the vessel, the left renal vein diameters in the projection of its compression by the superior mesenteric artery, in the area of the renal hilum, and between the superior mesenteric artery and the renal hilum were measured, and maximum blood flow velocity was determined at the site of its compression and in the area of the renal hilum [13,14].

Transvaginal duplex ultrasound (TVDUS) of the pelvic veins was performed using endovaginal micro-convex transducers with a frequency of 3–9 MHz and 3.5–10 MHz in the patient’s supine, half-sitting (with trunk raised to 45°) and half-standing positions [4,9,11]. The patency and diameter of the parametrial, uterine and gonadal veins (PVs, UVs and GVs, respectively) and IIV, as well as the presence and duration of reflux in these vessels were evaluated. The study of the IIV tributaries with identification of retrograde blood flow in them was not carried out due to inconclusiveness and low reproducibility of these parameters with DUS.

PVR was assessed using the previously developed classification, with types I (mild), II (moderate) and III (severe) PVR corresponding to the reflux duration of 1–2 s, 2.1–5 s and greater than 5 s (or with spontaneous reflux in PeVs without exercise load), accordingly [4].

SPECT of the pelvic veins with in vivo labeled red blood cells was performed for radionuclide assessment of PeVs and the degree of pelvic venous congestion (PVC). The Discovery NM/CT 670 (GE, St, Boston, MA, USA) and Philips Forte (Philips Healthcare, Best, The Netherlands) imaging systems were used [15]. The Perfotech solution for the in vivo labeling of RBCs was injected in the cubital vein in a dose of 2 mL, followed by 99mTc-pertechnetate injection 20 min later. SPECT was performed at 20 min post-injection using a 360° circular orbit rotation of gamma detector for obtaining images of the distribution of labeled RBCs in the pelvic veins. In a healthy individual, no or only negligible accumulation of a radiopharmaceutical (RPH) is identified in the OVs, PVs and UVs, while in PeVD patients the accumulation of RPH in PVs and UVs along with contrasting of GVs (most often of the left vein) is observed.

With the use of computer equipment for gamma camera, the radiation from RPH was measured in the regions of interest (in counts per second). For an objective assessment of venous congestion in the uterus and parametrium, the coefficient of pelvic venous congestion (C_PVC_) was calculated as the ratio of counts from “parametrial veins” to counts from the “common iliac vein”.

The activity of labeled RBCs in the common iliac vein is the most stable parameter. The activity of RBC-phosphate-pertechnetate complex in venous plexuses depends on the degree of their dilation, the presence of reflux and, thus, the degree of blood deposition in them. In healthy individuals, C_PVC_ does not exceed 0.5, while in PeV dilation and reflux, the labeled RBCs are accumulated in the dilated pelvic venous collecting pools, resulting in a significant increase in C_PVC_. The PVC severity was assessed by the C_PVC_ increase as follows: 0.5–1.0, grade I; 1.1–1.5, grade II, and ≥1.5, grade III.

### 2.2. Statistical Methods

Statistical analysis was performed using Microsoft Excel (Microsoft Corp., Redmond, WA, USA), Statistica 10 software (StatSoft, Tibko, Palo Alto, CA, USA) and VassarStats online calculator (open source online project; http://vassarstats.net/ (accessed on 22 November 2021)). We present continuous variables as either mean (SD) or median (interquartile range) as appropriate, and categorical variables as number and frequency (percentage of group). Comparisons were made using the Student’s *t*-test and Mann–Whitney test, where appropriate, for quantitative variables, and the Chi-square test for categorical variables. Differences were considered statistically significant at *p* < 0.05. To study linear relationships between different continuous variables with normal distribution, the Pearson’s correlation coefficient was used.

## 3. Results

The study groups did not differ significantly in demographic and clinical characteristics, except in the lower body mass index (BMI) and the rate of varicose veins of the lower extremities (VVLE) in symptomatic as compared to asymptomatic patients (Table 1). Significant differences between study groups were revealed using ultrasound and radionuclide studies.

### 3.1. Symptomatic Patients with PeVD

All patients of this group had moderate to severe PVR according to DUS (reflux duration in GVs and PVs of more than 2 s). Of them, 79% had type II reflux in UVs and 26% had a combined reflux in GVs, PVs, and UVs. The C_PVC_ values ranged from 1.3 to 2.8, which corresponded to grades II and III of pelvic venous congestion (Figure 2).

Therefore, ultrasound and radionuclide studies demonstrated a substantial impairment of venous outflow from pelvic veins, which was characterized by types II and III of PVR and C_PVC_ values indicating grades II and III of PVC. The pathological process involved PVs and UVs in 79% of these patients, and GVs, PVs and UVs in 26% of them.

### 3.2. Asymptomatic Patients with PeVD

In asymptomatic patients, a lower rate of moderate and severe PVR (5%) and a smaller accumulation of labeled RBCs in the pelvic venous plexuses were observed despite a significant increase in PeV diameters (Table 2, Figure 3). All patients showed a smaller accumulation of labeled RBCs in the parametrial and uterine veins. C_PVC_ did not exceed 0.7–0.9, which indicates grade I of PVC.

## 4. Discussion

Multiple pregnancies are among the main risk factors for the development of pelvic varicose veins (PVV) [16,17,18]. Pregnancy is accompanied with an increase in serum progesterone levels by tens of times, the volume of circulating blood by 15%, and the amount of blood flowing through the pelvic veins by tenfold. All of these factors can lead to PeV dilation and usually disappear after childbirth. In most women, the PeV status returns to normal, while in others the pelvic varicose veins persist and can constitute a basis for reflux development [19,20]. Subsequently, 60 to 76% of women with PeVD become symptomatic [4,21,22,23,24]. The reason for this phenomenon remains unclear [25,26].

At the same time, some authors state that it is the degree of pelvic vein dilation that determines the development of PeVD symptoms and signs [27,28,29]. According to others, an increase in the vein diameter is not a significant factor in the onset of clinical manifestations of PeVD, and the main mechanism contributing to the onset of CPP is the presence of reflux in PeVs and its duration [4,7,11,30,31]. Routine clinical practice shows that PeV dilation is quite a common observation; however, it is accompanied by PeVD symptoms in no more than a third of patients.

In the present study, a comparative analysis to identify relationships of clinical manifestations of PeVD with PeV diameters, PVR, and severity of blood deposition in the pelvic collecting pools was performed. PeV diameters were found to be similar in the symptomatic versus asymptomatic course of the disease (GVs: 7.7 ± 1.3 vs. 8.5 ± 0.5 mm, respectively, *p* = 0.56; PVs: 9.8 ± 0.9 vs. 9.5 ± 0.9 mm, respectively, *p* = 0.75; and UVs: 5.6 ± 0.2 vs. 5.5 ± 0.6 mm, respectively, *p* = 0.87). In addition, SPECT of the pelvic veins with in vivo labeled RBCs showed that with the same dilation of PeVs according to DUS data, the C_PVC_ was significantly higher in patients with PeVD symptoms than in asymptomatic patients (1.9 ± 0.4 vs. 0.7 ± 0.2, respectively, *p* = 0.008). Linear correlation analysis revealed a strong positive relationship (Pearson’s r = 0.78; *p* = 0.007) between the presence of pelvic pain and the PVR duration in patients with PeVD. On the contrary, the linear correlation between PeV diameters and the presence of clinical manifestations of PeVD was weak (r = 0.18), which indicates a minor influence of this factor. 

These findings suggest that in symptomatic patients, the pathological process involves a greater number of refluxing pelvic veins, as compared to asymptomatic patients (combined reflux in GVs, PVs and UVs in 26% vs. 5%, respectively, *p* = 0.007; combined reflux in PVs and UVs in 79% vs. 18%, respectively, *p* = 0.004). Symptomatic patients had PVR of types II or III (>2 s), while asymptomatic patients had PVR of type I (duration less than ≤ 2 s). The reflux duration was also longer in symptomatic versus asymptomatic patients (median and interquartile range: 4.0 [3.0; 5.0] vs. 1.0 [0; 2.0] s for GVs, *p* = 0.008; 4.0 [3.0; 5.0] vs. 1.1 [1.0; 2.0] s for PVs, *p* = 0.007; and 2.0 [2.0; 3.0] vs. 1.0 [1.0; 2.0] s for UVs, *p* = 0.04). These findings suggest that diameters or degree of dilation of the pelvic veins do not play a significant role in the development of clinical manifestations of PeVD and are not appropriate for its diagnosis. On the contrary, the PVR duration and its pattern in the pelvic veins should be considered as predictors of the development of symptomatic forms of PeVD.

This study had limitations due to the small number of patients and its open (non-randomized) retrospective nature. Despite the conclusive results of our research, future studies are warranted to confirm our findings in larger cohorts.

## 5. Conclusions

Findings from this study indicate that the leading factors in the development of symptomatic forms of PeVD are the duration of pelvic venous reflux, its prevalence in the pelvic veins, and blood deposition in the pelvic venous plexuses. The diameter of pelvic veins is not different in patients with symptomatic and asymptomatic PeVD, has no significant effect on the presence and severity of pelvic pain, and cannot be used as a diagnostic criterion. Duplex ultrasound serves as an accurate and objective method for assessing the status of pelvic veins, and its data can provide guidance for further examination of patients and confirm the necessity of additional radiation diagnostic tests.

## Figures and Tables

**Figure 1 diagnostics-12-00145-f001:**
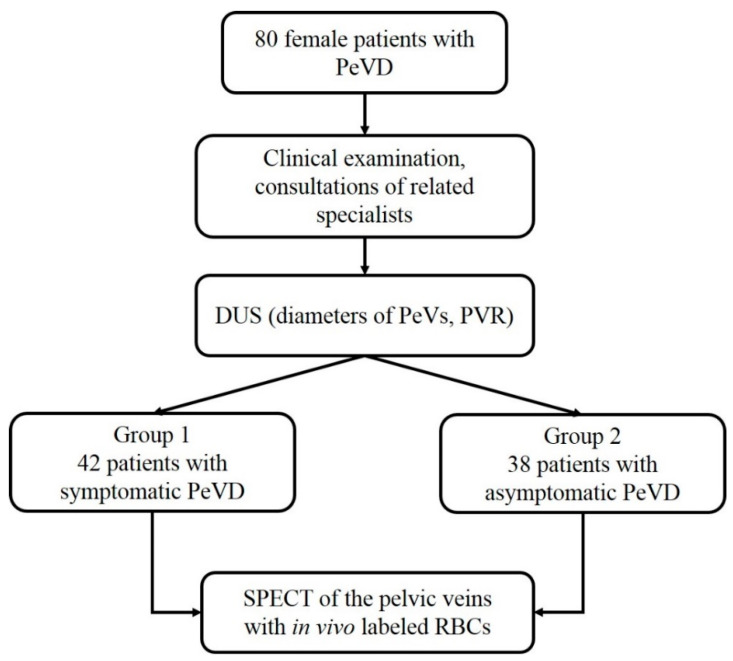
Study design. Abbreviations: PeVD, pelvic venous disorder; DUS, duplex ultrasonography; PeVs, pelvic veins; PVR, pelvic venous reflux; SPECT, single-photon emission computed tomography; RBCs, red blood cells.

**Figure 2 diagnostics-12-00145-f002:**
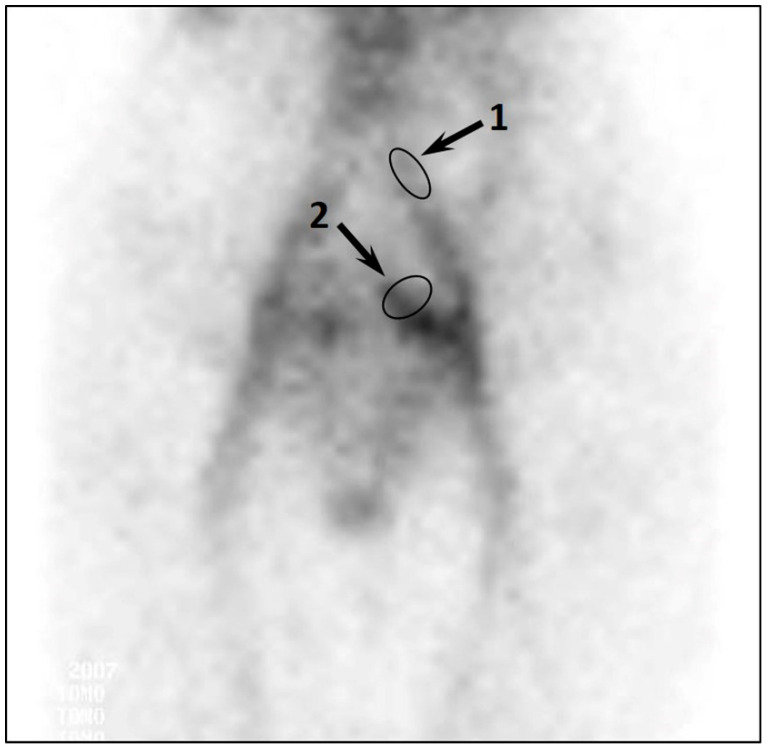
SPECT imaging of the pelvic veins of patient B. with symptomatic PeVD. A significant accumulation of labeled RBCs in the parametrial veins is observed. The arrows indicate the regions of interest: 1, “common iliac vein”; 2, “parametrial veins”. C_PVC_ = 1.6.

**Figure 3 diagnostics-12-00145-f003:**
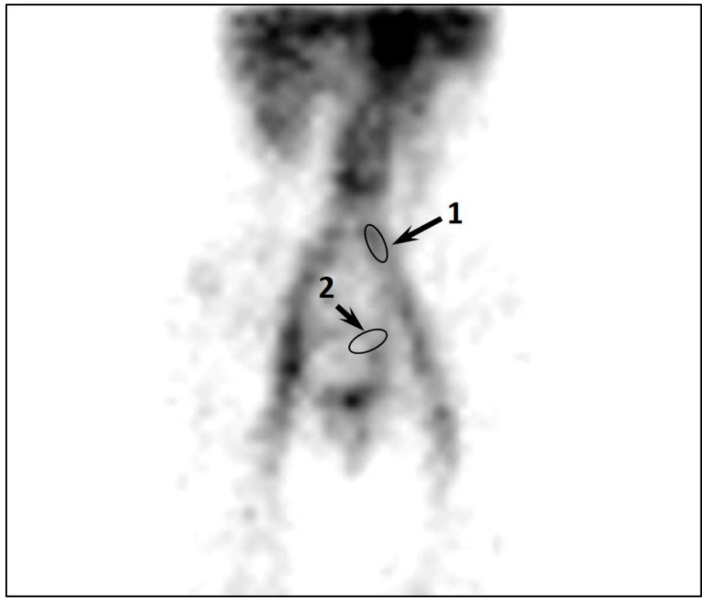
SPECT imaging of the pelvic veins of patient K. with asymptomatic PeVD. Low accumulation of labeled RBCs in the parametrial and uterine veins. The arrows indicate the regions of interest: 1, “common iliac vein; 2, “parametrial” veins. C_PVC_ = 0.8.

**Table 1 diagnostics-12-00145-t001:** Demographic and clinical characteristic of study participants (N = 80).

Variable	Symptoms of PeVD (n = 42)	No Symptoms of PeVD (n = 38)	*p* Value
Age, mean ± SD, years	32.5 ± 1.4	33.3 ± 1.2	0.6
BMI, mean ± SD, kg/m^2^	21.1 ± 1.2	24.5 ± 0.8	**0.02**
PeVD duration, mean ± SD, years	4.2 ± 2.3	3.3 ± 1.8	0.4
CPP, mean ± SD, VAS scores	7.4 ± 1.6	0	n.a.
Dyspareunia, mean ± SD, VAS scores	6.3 ± 0.9	0	n.a.
Heaviness in hypogastrium, n (%)	42 (100)	0	n.a.
Dysuria, n (%)	15 (36)	0	n.a.
Vulvar varicosities, n (%)	11 (26)	0	n.a.
VVLE, n (%)	6 (14)	17 (45)	**<0.01**
CVD of class 1 CEAP, n (%)	15 (36)	11 (29)	>0.01
Number of pregnancies, n	2–4	2–4	n.a.
Number of births, n	1–3	1–3	n.a.
Small uterine fibroids, n (%)	3 (7)	4 (11)	>0.01
Polycystic ovaries, n (%)	5 (12)	3 (8)	>0.01
Chronic colitis, n (%)	2 (5)	1 (3)	>0.01
Cholelithiasis, n (%)	4 (10)	6 (16)	>0.01

Significant differences are marked in bold. Abbreviations: SD, standard deviation; BMI, body mass index; PeVD, pelvic venous disorder; CPP, chronic pelvic pain; VAS, visual analogue scale; VVLE, varicose veins of the lower extremities; CVD, chronic venous disease; CEAP, Clinical-Etiological-Anatomical-Pathophysiological classification; n.a., not applicable.

**Table 2 diagnostics-12-00145-t002:** Results of ultrasound and radionuclide studies in asymptomatic and symptomatic patients with PeVD (N = 80).

Variable	DUS			SPECT		
	Without Symptoms, n = 38	With Symptoms, n = 42	*p* Value *	Without Symptoms, n = 38	With Symptoms, n = 42	*p* Value *
No reflux in GVs, n (%)	36 (95)	31 (74)	**0.03**	-	-	**-**
Diameter of non-refluxing GVs, mean ± SD, mm				-	-	-
Left	3.8 ± 0.3	4.2 ± 0.4	0.6			
Right	3.2 ± 0.2	3.5 ± 0.3	0.43			
Reflux in GVs, n (%)	2 (5)	11 (26)	**0.0**	-	-	-
Diameter of refluxing GVs, mean ± SD, mm				-	-	-
Left	8.5 ± 0.5	7.7 ± 1.3	0.5			
Right	no	6.8 ± 0.5	-			
Duration of reflux in GVs, M [IQR], s	1.0 [0; 2.0]	4.0 [3.0; 5.0]	**0.008**	-	-	-
Diameter of PVs, mean ± SD, mm	9.5 ± 0.9	9.8 ± 0.9	0.7	-	-	-
Reflux in PVs, n (%)	38 (100)	42 (100)	n.a.	-	-	-
Duration of reflux in PVs, M [IQR], s	1.5 [1.0; 2.0]	4.0 [3.0; 5.0]	**0.007**	-	-	-
Diameter of UVs, mean ± SD, mm	5.5 ± 0.6	5.6 ± 0.2	0.87	-	-	-
Reflux in UVs, n (%)	7 (18)	33 (79)	**0.004**	-	-	-
Duration of reflux in UVs, M [IQR], s	1.0 [1.0; 2.0]	2.0 [2.0; 3.0]	**0.04**	-	-	-
Visible GVs	-	-	-			-
Left GV, n (%)				0	8	
Right GV, n (%)				0	0	
Excessive accumulation of RPH in PVs and UVs	-	-	-	+	+++	-
C_PVC_	-	-	-	0.7 ± 0.2	1.9 ± 0.4	0.008

Significant differences are marked in bold. Abbreviations: GV, gonadal veins; SD, standard deviation; PV, parametrial veins; UV, uterine veins; PVR, pelvic venous reflux; RPH, radiopharmaceutical; Cpvc, coefficient of pelvic venous congestion; M, median; IQR, interquartile range; n.a., not applicable. * Calculated using the Student’s *t*-test or Mann–Whitney U-test as appropriate. “+” is a weak accumulation of RPH. “+++” is a significant accumulation of RPH.

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
