# Peer review of "Relationships of Pelvic Vein Diameter and Reflux with Clinical Manifestations of Pelvic Venous Disorder"

_diagnostics, 2022, doi:10.3390/diagnostics12010145_

Round 1

Reviewer 1 Report

This single-center study has evaluated the relationships of pelvic vein dilation and PVR with clinical manifestations of pelvic venous disorder in the two groups. 

There are contradictory opinions about the relationship between the degree of pelvic veins dilatation and clinical manifestations of pelvic disease. What emerged from this analysis was that the diameter of pelvic veins is not different in patients with symptomatic and asymptomatic pelvic disease, whereas the duration of pelvic venous reflux, its prevalence in the pelvic veins, and blood deposition in the pelvic venous plexuses could represent factors in the development of symptomatic forms of pelvic disease.

There are some limitations due to the small sample size and single-centre design, however it represents an interesting focus.

Please indicate the Power analysis of the study.

Please follow the standards for reporting p-values: 2 digits after dot for not significant and significant between 0.01- 0.05 and 3 digits for p-values <0.01

In table 2 please report median with interquartile range for variables that do not have normal distribution (for example duration of reflux) and use non-parametric test for comparison (e.g. Wilcoxon).

How many patients had may-Thurner Syndrom?

Why the Authors used SPECT than RM angiography for the pelvic vein study?

I do think this paper add new insights on this topic.

Author Response

This single-center study has evaluated the relationships of pelvic vein dilation and PVR with clinical manifestations of pelvic venous disorder in the two groups.

There are contradictory opinions about the relationship between the degree of pelvic veins dilatation and clinical manifestations of pelvic disease. What emerged from this analysis was that the diameter of pelvic veins is not different in patients with symptomatic and asymptomatic pelvic disease, whereas the duration of pelvic venous reflux, its prevalence in the pelvic veins, and blood deposition in the pelvic venous plexuses could represent factors in the development of symptomatic forms of pelvic disease.

Reply: Thank you very much for the appreciation of our work.

There are some limitations due to the small sample size and single-centre design, however it represents an interesting focus.

Reply: You are right. The sample size is small, which is explained by strict inclusion/exclusion criteria. Study limitations are described in the manuscript.

Please indicate the Power analysis of the study.

Reply: Almost two-third of consecutive patients did not meet the eligibility criteria because of the presence of other, non-venous disorders accompanied by chronic pelvic pain. The statistical power of the study had not been estimated due to specifics of the studied pathology and the absence of reference population data on the expected difference in pelvic venous diameters determining the pelvic pain occurrence. Therefore, our analysis should be considered as the pioneer one, and our findings warrant further studies on the larger number of patients. The respective notice had been added to the Discussion.

Please follow the standards for reporting p-values: 2 digits after dot for not significant and significant between 0.01- 0.05 and 3 digits for p-values <0.01

Reply: Corrected, accordingly.

In table 2 please report median with interquartile range for variables that do not have normal distribution (for example duration of reflux) and use non-parametric test for comparison (e.g. Wilcoxon).

Reply: The manuscript was amended accordingly with the median and IQR for non-normally distributed data (reflux duration). The use of Mann–Whitney U-test for these non-parametric variables was mentioned.

How many patients had May-Thurner Syndrome?

Reply: There were no patients with May-Thurner syndrome in the study cohort.

Why the Authors used SPECT than RM angiography for the pelvic vein study?

Reply: SPECT provides an opportunity to evaluate functional status of the pelvic veins, the degree of deposition of RP in the pelvic venous plexus. In addition, we have previously developed a radionuclide classification of the severity of pelvic venous congestion, based on the values of the TVP coefficient calculated with SPECT. MR venography provides information only on the structure and diameters of the veins, and for this purpose we used DUS.

I do think this paper add new insights on this topic.

Reply: Thank you for your appreciation of our study.

Reviewer 2 Report

Thank you for submitting the most scientific and important paper on the PCS.

I agree you findings and conclusion.

This paper contribute science a lot.

Gavrilov et.al demonstrate clear data importance of pelvic vein diameter and reflux for diagnosis of pelvic venous disorder. This paper might impact on newly developed “The Symptoms-Varices-Pathophysiology classification of pelvic venous disorders” which define pelvic varicose veins as “tortuous, dilated veins 5 mm or more in diameter around the ovary and uterus1). This study also demonstrates the first line diagnostic method is not CT or MRI but duplex scan which is less invasive and expensive.

1)Meissner MH, Khilnani NM, Labropoulos N, Gasparis AP, Gibson K, Greiner M, Learman LA, Atashroo D, Lurie F, Passman MA, Basile A, Lazarshvilli Z, Lohr J, Kim MD, Nicolini PH, Pabon-Ramos WM, Rosenblatt M. The Symptoms-Varices-Pathophysiology classification of pelvic venous disorders: A report of the American Vein & Lymphatic Society International Working Group on Pelvic Venous Disorders. Phlebology. 2021 Jun;36(5):342-360. doi: 10.1177/0268355521999559. Epub 2021 Apr 13. PMID: 33849310; PMCID: PMC8371031.

Author Response

Thank you for submitting the most scientific and important paper on the PCS.

I agree you findings and conclusion.

This paper contribute science a lot.

Reply: Thank you very much for your appreciation of our study.

Gavrilov et.al demonstrate clear data importance of pelvic vein diameter and reflux for diagnosis of pelvic venous disorder. This paper might impact on newly developed “The Symptoms-Varices-Pathophysiology classification of pelvic venous disorders” which define pelvic varicose veins as “tortuous, dilated veins 5 mm or more in diameter around the ovary and uterus1). This study also demonstrates the first line diagnostic method is not CT or MRI but duplex scan which is less invasive and expensive.